# Nutrient Intakes and Food Sources of Filipino Infants, Toddlers and Young Children are Inadequate: Findings from the National Nutrition Survey 2013

**DOI:** 10.3390/nu10111730

**Published:** 2018-11-11

**Authors:** Liya Denney, Imelda Angeles-Agdeppa, Mario V. Capanzana, Marvin B. Toledo, Juliana Donohue, Alicia Carriquiry

**Affiliations:** 1Nestlé Research, Vers-chez-les-Blanc, Lausanne 1000, Switzerland; Juliana.Donohue@rd.nestle.com; 2Food and Nutrition Research Institute, Department of Science and Technology, Taguig City 1631, Philippines; iangelesagdeppa@yahoo.com.ph (I.A.-A.); mar_v_c@yahoo.com (M.V.C.); marvinbtoledo@gmail.com (M.B.T.); 3Iowa State University, Ames, IA 50011, USA; alicia@iastate.edu

**Keywords:** usual nutrient intakes, food sources, infants, young children, The Philippines

## Abstract

Comprehensive assessment of nutrient intakes and food sources of nutrients in Filipino children under 5 years old are lacking. We studied energy and nutrient intakes and food sources in 4218 children aged 6–59.9 months using two 24-h dietary recalls. Usual energy and nutrient intakes were estimated using the PC-SIDE program. Reported foods and beverages were assigned to one of 85 food groups. Percentage contribution of each food group to nutrient intake was calculated. The results showed that the intake of total fat as a percentage of energy and of most micronutrients were highly inadequate. The prevalence of inadequate nutrient intakes, defined as the percent of children with intakes less than the estimated average requirements (EAR) ranged from 60–90% for iron, calcium, vitamin C, and zinc and ranged from 30–50% for others such as vitamin A, thiamine, riboflavin, niacin, vitamin B6, and phosphorus. The diets of these children were composed of limited foods, namely a large amount of refined rice and other low-nutrient-dense foods (cookies and sugar), while vegetables, fruits, meats, and eggs made little contribution to daily energy or nutrients. These findings provide direction to health professionals developing food-based recommendations and strategies to tackle the shortfalls in the diet of this population.

## 1. Introduction

Nutrition in early life is crucially important for children to grow and develop into healthy adults. Children can reach their growth and development potential if their nutritional needs are met in a healthy environment [1,2,3]. Unfortunately, the Philippines is one of the countries in the world where a significant number of children remain malnourished despite the economic growth and development in the country over the past decades [4]. Data from the 2013 National Nutrition Survey (2013 NNS) in the Philippines reported that among children under 5 years old, the prevalence of malnutrition measured by underweight and stunting was 20% and 30%, respectively, and the prevalence starts to increase at 6–11 months [5]. In addition, the prevalence of anemia among these children was 13.8%. Infants aged 6–11 months had the highest prevalence of anemia at 40.5%, and the prevalence of vitamin A deficiency was 27.9% [6].

Compared with other population groups, less is known about the dietary status of Filipino infants and toddlers, especially how subgroups of different ages vary in their nutrient intakes and food consumption during this critical growth period. There have been a number of studies focusing on certain nutrition topics and geographic areas in the Philippines. For example, macronutrient and selected vitamin intakes from complementary foods were investigated in infants and toddlers in the Cebu province. That study found that with the exception of protein, intakes of energy and vitamins for all age groups were below the World Health Organization estimated needs and desired nutrient densities from complementary foods [7]. Rohner and colleagues reported infant and young child feeding practices and their associations with stunting, anemia and deficiencies of iron and vitamin A in urban Philippines [8]. Other studies included determinants of anemia among preschool children from rural villages in Cebu and the use of dietary diversity scores as an indicator of micronutrient intakes in children aged 24–71 months [9,10,11].

Up to now, there has been no comprehensive assessment of nutrient intakes and food sources of nutrients in a nationally representative sample of infants and young children in the Philippines and surrounding countries in Southeast Asia, where the prevalence of malnutrition is also high [12]. Such a study is needed to demonstrate the proportion of the population that are not meeting dietary recommendations, as well as to provide insight on the relative contribution and importance of specific foods and food groups consumed in the population. This knowledge can provide a basis for population-level estimates and assist health professionals to form targeted measures to improve shortfalls. The purpose of this study was to use data from the NNS 2013 to describe the dietary intakes of infants, toddlers and young children aged from 6 months to 5 years including (1) the prevalence of inadequate nutrient intakes, (2) food groups consumed and (3) contribution of food groups to energy and nutrients.

## 2. Methods

### 2.1. Study Population

Data from 4218 infants, toddlers and young children aged 6–59.9 months who participated in the 2013 NNS were used in the current analyses. The 2013 NNS is a cross-sectional, population-based survey that characterizes the health and nutritional status of the Filipino population. The survey used a stratified three-stage sampling system drawn to represent all 17 regions and 80 provinces of the country covering both urban and rural areas. A total of 35,825 Filipino households were sampled with a response rate of 91%. The Ethics Committee of Food Nutrition Research Institute (FNRI) approved the survey protocol and data collection instruments. All surveyed households provided informed consent prior to participation.

### 2.2. Data Collection

Two 24-h dietary recalls were conducted face-to-face with the parent or caregiver of each child. Trained registered dietitians carried out the interviews during household visits using a structured questionnaire. To estimate the day-to-day variance component in energy and nutrient intake required for usual intake analysis, the first 24-h dietary recall was collected for all children and a second 24-h dietary recall was repeated in 50% of randomly selected households on a non-consecutive day. The second 24-h dietary recall was typically collected 2 days after the first 24-h recall. The interviewer asked about all foods and beverages that the child consumed on the previous day. The amount of each food item or beverage was estimated using common household measurements such as cups, tablespoons, by size or number of pieces. The information was then converted to grams using a portion to weight list for common foods compiled by FNRI or through actual weighing of the food samples.

Data on family economic status, weight and height of the children were taken from the same NNS 2013 reports. Briefly, family economic status was assessed by wealth quintiles defined as poorest, poor, middle, rich, and richest. The wealth status was defined by household possession of vehicles, gadgets and appliances [13]. The World Health Organization-Child Growth Standards were used to assess the nutritional status of children 0–60 months based on weight and height measurements [5,14].

### 2.3. Data Processing

Food records were entered and estimated energy and nutrient intakes were processed with a computer system called Individual Dietary Evaluation System. This system contains the data of an updated Filipino Food Composition Tables (FCT) created for this study. The updated FCT contains 27 nutrients from 1359 foods. About half of the data (47%) were from the United Sates Department of Agriculture (USDA) national nutrient database and 39% of the data from the original Filipino FCT compiled by FNRI [15]. The rest of the data were sourced from the food composition database of Association of Southeast Asian Nations and other Asian countries such as Japan (8%) and information from food labels (6%). All imported data were adapted according to the International Network of Food Data Systems guidelines Food Agriculture Organization [16].

Quality control of the dietary intake data was conducted in two steps. In the first step, the foods reported by a participant were reviewed and information including coding and quantity reported. In the second step, energy and nutrient intakes were reviewed to identify implausible values by using the method described by Lopez-Olmedo and colleagues [17]. The estimated energy requirement (EER) was calculated for each individual by using the equations for maintenance of body weight from the Institute of Medicine based on age, gender, weight, height, and physical activity information [18]. We assumed a sedentary physical activity level for this study population. For implausible energy intake, the ratio of daily energy intake to EER was calculated for each person and transformed to a logarithmic scale to remove outliers below −3 SDs and above +3 SDs. For implausible micronutrient intakes, excessive intakes were defined as those that exceeded 1.5 times the 99^th^ percentile of the observed intake distribution of the nutrient in the corresponding sex and age group. Intakes above the upper limit were substituted by a random value generated from a uniform distribution in the intervals with the lower bound equal to the 95^th^ percentile of the observed intake and an upper bound equal to 1.5 times the 99^th^ percentile [17]. After data editing and processing, 47 individuals were excluded from the analysis for energy intake.

Breast milk consumption was estimated based on the child’s age in months and total amount of other milks consumed (such as infant formula and cow’s milk) using the information from published literature [19,20]. For infants aged 6 to 11.9 months fed human milk as the sole milk source, the amount of human milk was assumed to be 600 mL/day; for partially breastfed infants, the amount of human milk was computed as 600 mL/day minus the amount of infant formula/other milks consumed. For breastfed toddlers aged 12–23 months, the amount of human milk was computed as 529 mL/day minus the amount of infant formula/other milks consumed [21]. For toddlers aged 24–35.9 months, the amount of human milk was estimated as 59 mL per feeding occasion.

To investigate food groups consumed and food sources of energy and nutrients, a list of 85 food groups was created in a similar format from previous dietary intake studies in young children in the USA [22,23] (Table 1). Two trained Filipino nutritionists and a nutrition scientist at Nestlé Research adjusted food groups to incorporate local food culture and habits. Data analyses for food groups consumed and sources of energy and nutrients were based on the first 24-h dietary recall. All foods and beverages reported were assigned to one of the 85 food groups. Fortified milk powder produced for toddlers and young children in the Philippines is called toddler and preschooler formula in this study.

### 2.4. Statistical Analysis

Mean and usual intake distributions of energy and nutrients were estimated using the software developed by Iowa State University, PC-SIDE version 1.02 (Iowa State University, Ames, IA, USA) and within-person variation of nutrient intake was accounted for across days. This program estimates distributions (in percentiles) of usual nutrient intake by removing the effect of day-to-day (intra-person) variability in intake from daily intakes, and then calculating the proportion below the estimated average requirements (EAR) defined by the Philippine Dietary Reference Intakes 2015 [24]. Hence, the prevalence of inadequacy in the population is estimated as the proportion of individuals with usual intakes below the EAR [25].

Acceptable Macronutrient Distribution Ranges (AMDR) were used to evaluate carbohydrates, total fat, and protein intakes as a percentage of energy. The proportions of inadequate and excessive intakes were classified as less than the AMDR lower range and greater than AMDR upper range, respectively. For the prevalence of inadequate intakes of iron, a probability approach was used [26]. First, the risk of inadequacy of each individual was computed and then the prevalence of inadequate iron intake was calculated, which is the average risk of inadequacy. Calculations for summary statistics were carried out using STATA version 13 (StataCorp, College Station, TX, USA).

Food group consumption was expressed as the percentage of children who consumed specific food groups at least once on the first 24-h dietary recall regardless of the amount consumed. This method has been used in our previous studies [22,27]. The weighted percentage contribution of each food group for selected key nutrients was calculated by adding the amount of a given nutrient provided by each food group for all individuals and dividing by the total intake of that nutrient consumed by all individuals from all foods and beverages. In order to understand food consumption in detail and how it changes with age, data for nutrient intakes, food groups and sources of nutrients are presented for four age groups: 6–11 months, 12–23.9 months, 24–35.9 months and 36–59.9 months.

## 3. Results

### 3.1. Demographic Charateristics of the Study Population

The characteristics of children, mothers and family wealth status are shown in Table 2. Approximately 50% of the participants were boys and 50% were girls. Fifty-six percent of the children were from urban regions and the rest were from rural regions. Half of the families (50.7%) were classified as poor and poorest. Among the mothers, nearly 77% had an education above high school and 71.2% did not have an occupation outside the home. The majority of the mothers were married or lived with their partner (89.7%). The prevalence of underweight and stunting among the children included in this study were 21.1% and 29.3% respectively while prevalence of wasting and overweight were 5.7% and 3.9% respectively.

### 3.2. Nutrient Intakes and Food Sources of Infants (6–11.9 Month Olds)

The mean energy intake of 688 kcal/day was 7.8% higher than the estimated EER (mean ± SE): 638 ± 6 kcal/day. Fifteen percent of the infants had percentage of energy from total fat below while 36% exceeded the AMDR in this age group (Table 3). High prevalence of inadequate intakes were found for vitamin A (84%), iron (76%) and zinc (63%). Considerable high prevalence of inadequate intakes were found for protein (43%), thiamine (52%), riboflavin (45%) and niacin (58%). Mean intakes of vitamin C, vitamin B6, folate and magnesium were above the AIs while mean intakes of vitamin E, phosphorus, vitamin D and potassium were far below the AIs. (Table 2). Mean sodium intake (303 mg/d) was above the AI (200 mg/d).

Consumption of the top-20 most consumed food groups and their contribution to energy and selected nutrient intakes are presented in Table 4. Refined rice, human milk, infant formula, cookies and cow’s milk were the top-5 foods most consumed followed by infant cereal, vegetables, grain-based mixed dishes, table sugar and fish (Table 4). Fruits were only consumed by 5.5% of the infants.

Human milk, infant formula, rice, cow’s milk and infant cereal were the top-5 sources of energy and most of the selected key nutrients (Table 4). Besides the top-5 foods, the proportion of infants consuming other food groups was low and their contribution to nutrient intakes was small except for toddler/preschooler formula. Although toddler/preschooler formula was only consumed by 3.7% of the infants, it was among the top four or five sources of total fat, vitamin A, vitamin C, calcium and iron, indicating many other food groups contributed very little to nutrient intakes (Table 4).

### 3.3. Nutrient Intakes and Food Sources of Younger Toddlers (12–23.9 Month Olds)

Mean energy intake of 771 kcal/day just met the estimated EER (mean ± SE): 784 ± 6 kcal/day. High prevalence of inadequate intakes were found for iron (78%), folate (68%), vitamin B6 (61%), vitamin A (60%), and calcium (62%). Inadequacy was also found for thiamine (59%), riboflavin (42%), niacin (57%), vitamin B12 (53), phosphorus (57%), and zinc (52%) (Table 5). Mean intakes of vitamin E, vitamin D and potassium were all far below the AIs but mean sodium intake (517 mg/d) was about twofold higher than the AI (225 mg/d).

Refined rice, human milk, cow’s milk, fish, and vegetables were the top-5 foods most consumed, although vegetables were only consumed by 28% of the children (Table 6). The next five food groups were cookies, toddler/preschooler formula, bread, noodles, and eggs.

Among the younger toddlers, refined rice, human milk, cow’s milk, toddler/preschooler formula, and cookies were among the top-5 sources of energy. The above first four foods together with sugar-sweetened beverages were the top-5 sources of selected key nutrients (Table 6). Briefly, refined rice was the first source of protein and the third or fourth source of thiamine, riboflavin, iron, zinc and calcium. Cow’s milk was the first source of riboflavin, vitamin A and calcium. Toddler/preschooler formula was the first source of thiamine, vitamin C, iron and zinc. Sugar-sweetened beverages were the third to fifth sources of thiamine, riboflavin, vitamin A, calcium, iron and zinc (Table 6). Like infants, other food groups were consumed by small percentages of the children and contributed little to nutrient intakes overall.

### 3.4. Nutrient Intakes and Food Sources of Older Toddlers (24–35.9 Month Olds)

Mean energy intake, 838 kcal/day, was about 13% below the estimated EER (mean ± SE): 962 ± 7 kcal/day. A high proportion, 58% of the older toddlers, consumed a low percentage of energy from total fat (Table 7). Similar to younger toddlers but with slightly lower proportions, a high prevalence of inadequate intakes were found for almost all of the vitamins and minerals: iron (75%), calcium (66%), folate (63%), thiamine (46%), phosphorus (44%), zinc (46%), vitamin A (41%), vitamin B6 (40%), vitamin C (35%), riboflavin (35%), niacin (30%) and vitamin B12 (29%). Mean intakes of vitamin E, vitamin D and potassium were all far below the AIs while the mean sodium intake (646 mg/d) was more than twofold higher than the AI (225 mg/d) (Table 7).

Among the older toddlers, refined rice, fish, vegetables, cow’s milk and sugar-sweetened beverages were the top-five foods most consumed followed by table sugar, bread, noodles, eggs and cookies (Table 8). Fruits were consumed by 17% and toddler/preschooler formula by 15% of the children.

Refined rice, cow’s milk, toddler/preschooler formula, bread, and noodles were top-5 sources of energy. For nutrients, refined rice, cow’s milk, sugar-sweetened beverages and toddler/preschooler formula and pork were the top-5 sources (Table 8). Notably, sugar-sweetened beverages were consumed more in this age group (33%) and became top sources of all the selected nutrients (Table 8). Other food groups such as vegetables, bread, noodles and fruits were among top-5 sources for some nutrients. For example, vegetables were the fifth sources of vitamins A and C and fruits were the third source of vitamin C (Table 8).

### 3.5. Nutrient Intakes and Food Sources of Young Children (36–59.9 Month Olds)

Mean energy intake, 997 kcal/day, was 11% below the estimated EER (mean ± SE): 1119 ± 2 kcal/day. Twenty-seven percent of the children consumed a low proportion of energy from total fat (Table 9). High prevalence of inadequate intakes were found for iron (90%), calcium (84%), vitamin C (60%), folate (72%), zinc (47%), thiamine (43%) riboflavin (43%) and vitamin A (43%). Mean intakes of vitamin E, vitamin D and potassium were all far below the AIs while mean sodium intake (740 mg/d) was more than twofold higher than the AI (300 mg/d) (Table 9).

In this age group, refined rice, fish, vegetables, sugar-sweetened beverages, and cow’s milk were the top-5 foods most consumed followed by table sugar, bread, eggs, noodles, and pork (Table 10). For the sources of energy and nutrients, refined rice, cow’s milk, bread, sugar-sweetened beverages, and noodles were the top-5 sources of energy with rice alone, contributing 41%. Rice was also the first source of protein, thiamine, iron and zinc. Sugar-sweetened beverages were the first source of vitamin C and the second source of thiamine, riboflavin, vitamin A, and iron. Cow’s milk was the first to fifth source of all nutrients except iron (Table 10). Other foods among the top sources of nutrients were bread, pork, chicken, and fruits. Although fruits were only consumed by 17% of the children, they were the second highest source of vitamin C.

## 4. Discussion

Our analyses focused on nutrient intakes from food and beverages only to provide important information to inform compliance with food-based dietary guidance or to develop interventions to better meet nutrient needs. The data provide an overview of dietary patterns among Filipino infants, toddlers and young children and demonstrate that the diet of this population group is far from optimal. To our knowledge, this is the first study to provide current and comprehensive estimates of usual intakes of nutrients and food sources of key nutrients in children under 5 years old in the Philippines. In addition, the finding of this study could also provide insights to plan similar studies in other countries in Southeast Asia, where the diets of children are also largely based on rice and malnutrition remains a pressing issue.

### 4.1. Inadequate Nutrient Intakes

We examined the prevalence of inadequate nutrient intakes relative to the EAR when available and percentile distribution and mean intakes were presented for the rest of the nutrients. The results demonstrated that a high prevalence of inadequate intakes were found for all nutrients evaluated except for selenium. The prevalence of inadequate intakes ranged from 60% to 90% for iron, calcium, vitamin C, folate and zinc and from 30% to 50% for other nutrients. In addition, no age group met the AIs for vitamin E, vitamin D or potassium, implying a high risk of inadequacy for these nutrients as well. Our results are in line with previous studies and surveys, in which large shortfalls in nutrient intakes were reported in the Philippines, although there were differences in the methodology of nutrient intake assessment [7,28,29].

The period from birth to age five is a critical window for optimal growth and development [3]. Our results suggest that there are large shortfalls in the diet of this population and this could be one of the factors contributing to the high prevalence of underweight, stunting and nutrition deficiencies [11,30].

### 4.2. Low Food Variety and Diets Lack Nutrient-Dense Foods

As an infant grows and develops, it experiences physiological shifts in nutrient and energy requirements that can no longer be supported by breast milk alone. Therefore, complementary foods with relative high energy and nutrient density and of good variety must be provided [31]. In this population, we found that the majority of the children consumed few foods, many of which were low in nutrients, such as rice, cookies and sugar.

Breast milk was consumed by 60% of infants (6–11.9 months) and 37% of toddlers (12–23.9 months). This is suboptimal based on breastfeeding guidelines of the World Health Organization [32] but these rates are higher than for infants and toddlers in studies conducted in China and the United States [27,33]. Breast milk was indeed a main source of all nutrients in infants 6–11.9 months and toddlers 12–23.9 months in this study. Other milk sources including infant formula and toddler/preschooler formula in infants and toddlers and cow’s milk in young children also played important roles in providing nutrients. Another important finding is the role of fish in the diet. Fish was the second most consumed food among toddlers 24–35.9 months and young children 36–59.9 months, and was an important food source for a number of key nutrients including protein, calcium, iron, zinc, vitamin A, and riboflavin.

However, the fact that refined rice was the first source of energy and the top source of many key nutrients in all age groups indicates that besides milk and fish, other nutrient-rich foods are missing from the diet. Indeed, we found that overall vegetables, fruit, and meats were only consumed by a small proportion of children and contributed little to nutrient intake. For example, fruits were only consumed by 5% of the infants and 14 to 17% of toddlers and young children. Vegetable consumption was as low as 11% among infants, in other words, only 1 in 10 of the infants consumed vegetables. Although older children consumed more vegetables, the quantity was less than 20 grams per capita per day in young children aged 36–59.9 months. In addition, other nutrient-rich foods such as meats, eggs and infant cereals were not commonly consumed either. About 10 to 25% of the children consumed eggs and meats. Infant cereal was consumed by 14% of the infants. Infant cereal is fortified with iron and other nutrients, a complementary food recommended by nutrition and medical organizations in many countries [3,34]. In fact, our data show that infant cereal was among the top food sources of almost all nutrients, although it was only consumed by a small proportion of the children.

Another unique observation is that sugar-sweetened beverages were among the top sources of nutrients in the diets of these children. In Western countries, sugar-sweetened beverages are discouraged by health professionals due to high sugar content [3]. In the Philippines, sugar-sweetened beverages are affordable and are among commonly consumed foods. Therefore, sugar-sweetened beverages are often used as carriers for fortification [35,36,37]. This makes sugar-sweetened beverages sources of micronutrients. We could not evaluate added sugar intake in this study because the data are not currently available in the Philippine FCT database.

### 4.3. Strengths and Limitations

This study provides one of the largest, population-based sources of data on usual energy and nutrient intakes from children aged 6 months up to 59.9 months. Data are presented by short age intervals to capture shifts in dietary patterns during this period of rapid growth and development. This study has several limitations. First, we examined intake on a given day, therefore, we may have underestimated the consumption of foods that are not consumed on a daily basis. Second, this study relied on mother or caregiver reports of child intake. The participants may have over- or under-estimated their child’s consumption during the recall [38]. However, because of the large sample size with corresponding survey weights applied in all the datasets, these limitations could be overshadowed and could represent valuable national information. Third, because the purpose of this study was to estimate nutrient intakes from food and beverages only, the use of dietary supplements were not included. This could lead to underestimation of total daily intakes of micronutrients in these children. A further study that assesses nutrient intakes from foods, beverages and dietary supplements is warranted. Lastly, about half of the data in the food composition database was built by adopting data from the National Nutrient Database of United States Department of Agriculture and from some other nearby countries. Although efforts were made to select the foods that could match the equivalent foods in the Philippines, food composition of foods from other countries may not reflect the foods in the Philippines. This could also induce over- or underestimation of nutrient intakes in this population.

## 5. Conclusions

This study provided important insights into the dietary patterns among Filipino children aged 6–59.9 months. The results showed that the intakes of total fat as a percentage of energy and most of micronutrients were highly inadequate. This can be explained largely by the low variety and low nutritional quality of foods consumed in this population. The diets of these children were made of few foods, namely the majority of energy and nutrients come from refined rice and other low nutrient-dense foods (cookies and sugar), while vegetables, fruits, meats and eggs made little contribution. The findings from this study should provide direction to healthcare professionals developing food-based recommendations and may contribute to intervention strategies to tackle the shortfalls in the diet of this population. Further studies to understand factors influencing the dietary intake of this population, such as socioeconomic status of families, food consumption habits and food access in different resigns are underway. In addition, the findings and insights obtained from this study could be shared with the nutrition researchers and healthcare professionals in other countries in Southeast Asia, where malnutrition of children is prevalent.

## Figures and Tables

**Table 1 nutrients-10-01730-t001:** Food group classification.

Milk & Milk Products	Vegetables	Cookies
Infant formula (powdered)	Dark green leafy vegetables	Biscuits
Toddler/preschooler formula (powdered)	Spinach	Sweet breads
Milk (liquid and powdered)	Broccoli	Cakes
Cheese	Cabbage, green	Ice cream/popsicles
Yoghurt	local leafy/petioles/salad vegetables	Candy
**Meats/poultry/fish/beans**	Deep yellow vegetables	Sugar
Beef	Carrot	Syrup
Carabeef	Sweet potato, yellow	Preserves/jams/jellies
Pork	Cassava, yellow	Native snacks
Goat/lamb	Squash fruit	Savory snacks
Chicken	Squash, summer fruit	**Sugar sweetened beverages**
Duck	Other vegetables	Fruit-based beverages
Sausages	Sweet potato	Concentrated fruit juice drinks
Luncheon meats	Potato	Powdered fruit flavored drinks
Cold cuts	**Fruits & 100% fruit juice**	Ready to drink fruit flavored drinks
Fish	Fruits, fresh	Soft drinks (carbonated Cola)
Eggs	Apple	Chocolate/chocolate flavor beverages (powdered)
Beans/nuts	Avocado	Other sweetened beverages (liquid and powdered)
**Grains & grain products**	Banana	**Mixed dishes**
Cereal	Mango	Meat-based mixed dishes
Bread	Melon	Bean-based mixed dishes
Crackers	Citrus fruits	Grain-based mixed dishes
Pancakes/waffles/French toast	Cherries/berries	Soups
Rice	Papaya	**Other**
Pasta	Food fruits, canned	Fats/oils
Noodles	100% fruit juice (lemon, mango, apple and pineapple)	Condiments/sauces/other seasonings
Corn grits	**Sweets & snacks**	
Cornmeal	Sweet bakery products	

**Table 2 nutrients-10-01730-t002:** Characteristics of the study population.

		*n*	%
**Gender**	Boy	2143	50.3
	Girl	2121	49.7
**Age**	6–11.9 month	362	8.5
	12–23.9 month	734	17.2
	24–35.9 month	741	17.4
	36–59.9 months	2427	56.9
**Wealth quintiles**	Poorest	1224	29.5
	Poor	880	21.2
	Middle	820	19.8
	Rich	684	16.5
	Richest	542	13.1
**Region**	Urban	2378	55.8
	Rural	1886	44.2
**Underweight**	6–11.9 month	54	1.3
	12–23.9 month	128	3.0
	24–35.9 month	160	3.8
	36–59.9 months	543	12.9
	All	885	21.1
**Stunted**	6–11.9 month	52	1.2
	12–23.9 month	198	4.7
	24–35.9 month	235	5.6
	36–59.9 months	740	17.7
	All	1225	29.3
**Wasted**	6–11.9 month	43	1.0
	12–23.9 month	50	1.2
	24–35.9 month	40	1.0
	36–59.9 months	105	2.5
	All	239	5.7
**Overweight**	6–11.9 month	14	0.4
	12–23.9 month	29	0.9
	24–35.9 month	29	0.9
	36–59.9 months	62	1.8
	All	134	3.9
**Mother’s marital status**	Single	227	6.7
	Married	2325	68.7
	Live-in	712	21.1
	Separated/divorced/widowed	117	3.4
	Unknown	2	0.1
**Mother’s education**	No grade completed	52	1.2
	Elementary level	828	21.7
	High school level	1655	49.1
	Vocational level	158	4.9
	College level	794	22.9
	Others	7	0.02
**Mother’s current work**	No occupation	2426	71.2
	Student	24	0.01
	With job/business	920	28.1

**Table 3 nutrients-10-01730-t003:** Usual energy and nutrient intakes from food and beverages for Filipino infants aged 6–11.9 months (*n* = 350).

	Dietary Reference Intakes ^1^			Mean/Median Intake Percentiles						Inadequate/Excessive Reported Intake	
Nutrients	EAR/AMDR	AI/RNI	UL	10th	25th	Median	Mean ± SE	75th	90th	% < EAR/AMDR	%>AMDR/>UL
**Macronutrients**											
Energy intake (kcal/day)	638 (EER)			443	497	610	688 ± 14	815	1066	-	-
Total fat (g/d)	-	-	-	18.9	21.6	24.7	24.5 ± 0.5	30.7	40.8	-	-
Saturated fat (g/d)	-	-	-	1.1	3.5	8.4	7.3 ± 0.2	10.4	11.8	-	-
Protein (g/d)	13.5	-	-	14.8	11	14.8	18.6 ± 0.6	23	32.9	43	-
Carbohydrate (g/d)	-	-	-	53	63	80	90.6 ± 2	109	145	-	-
Total sugars (g/d)	-	-	-	6.1	26.1	31.7	34.9 ± 1.2	42.4	60.8	-	-
Dietary fiber (g/d)	-	-	-	0.1	0.3	0.7	1.3 ± 0.1	1.6	2.7	-	-
**As percentage of total energy**											
Total Fat (%)	30–40	-	-	27.5	33.2	37.9	36.7 ± 0.4	41.5	44.1	15	36
Protein (%)	8–15	-	-	8.1	8.5	9.6	10.3 ± 0.1	11.6	13.5	7	5
Carbohydrate (%)	45–62	-	-	45.3	48.6	51.7	52.9 ± 0.4	55.9	61.4	9	9
**Antioxidants**											
Vitamin C (mg/d)	-	-		21.9	26.2	34.8	54.6 ± 2.3	72	120.5	-	-
Vitamin E (mg/d)	-	4	-	0.3	0.5	1.1	2.8 ± 0.2	3.6	8.4	-	-
**B vitamins**											
Thiamine (mg/d)	0.3	-	-	0.1	0.1	0.3	0.5 ± 0.03	0.7	1.2	52	-
Riboflavin (mg/d)	0.3	-	-	0.1	0.1	0.4	0.8 ± 0.1	1.3	2	45	-
Niacin (mg/d)	3.5	-	-	1	1.4	2.8	3.9 ± 0.2	5.5	8.4	58	-
Vitamin B6 (mg/d)	-	0.25	-	0.1	0.1	0.2	0.3 ± 0.03	0.5	1	-	-
Folate (DFE µg/d)	-	75	-	26	37	63	95.6 ± 5	122	217	-	-
Vitamin B12 (µg/d)	-	-	-	0.2	0.4	0.6	1.1 ± 0.1	1.4	2.7	-	-
**Bone-related nutrients**											
Calcium (mg/d)	-	-		153	182	303	522 ± 27	727	1170	-	-
Phosphorus (mg/d)	-	-		106	135	234	364 ± 18	485	804	-	-
Magnesium (mg/d)	-	50	-	17	24	37	53.1 ± 2.6	65	112	-	-
Vitamin D (µg/d)	-	5	25	0.5	0.9	1.6	4.3 ± 0.3	5.4	12	-	-
**Other micronutrients**											
Vitamin A (µg RE/d)	190	-	600	3.8	25.6	51.3	126 ± 12	126.3	269	84	
Iron (mg/d)	8.4	-	-	1.6	2.2	3.7	6.5 ± 0.4	8.1	15.5	76	-
Zinc (mg/d)	2.65	-	5	0.9	1.3	1.9	3 ± 0.1	3.8	7	63	18
Sodium (mg/d)	-	200	-	126	167	245	303 ± 10	379	567	-	-
Potassium (mg/d)	-	700		245	338	435	579 ± 22	694	1149	-	-
Selenium (µg/d)	7.75	-	60	6.6	12.2	15.7	18.6 ± 0.7	22.4	33.5	12	1

^1^ Philippine Dietary Reference Intakes 2015. EAR, estimated average requirements. AMDR, acceptable macro-nutriment distribution range. AI, adequate intake. RNI, reference nutrient intake. UL, tolerable upper intake level. DFE, dietary folate equivalents. RE, retinol equivalents.

**Table 4 nutrients-10-01730-t004:** Top-20 most consumed food groups and their contribution to energy and selected nutrient intakes among infants aged 6–11 months (*n* = 350) in the Philippines.

				Percent Contribution to Total Daily Intake (Ranking of Top 5 Food Sources of Nutrients)											
				Macronutrients				Vitamins					Minerals		
Rank	Food Groups	% of Children	Mean Intake (SE) Per Capita (g)	Energy	Carbohydrate	Protein	Total Fat	Thiamine	Riboflavin	Vitamin A	Vitamin C	Calcium		Iron	Zinc
1	Rice	70.0	35.2 (7.6)	**11.7 (3)**	**19.9 (3)**	**8.9 (4)**	0.5	**5.9 (5)**	**1.8 (5)**	0	0	1.3		**3.7 (4)**	**6.6 (3)**
2	Human milk	60.0	336.1 (5.4)	**33.9 (1)**	**28.5 (1)**	**25.8 (2)**	**43.9 (1)**	**7.6 (3)**	**4.5 (4)**	**6.7 (3)**	**26.3 (2)**	**17.3 (3)**		**16.9 (3)**	**21.2 (2)**
3	Infant formula	35.1	38.6 (3.7)	**27.6 (2)**	**24.8 (2)**	**30.5 (1)**	**31.5 (2)**	**55.0 (1)**	**52.9 (1)**	**60.6 (1)**	**56.3 (1)**	**50.2 (1)**		**44.5 (1)**	**44.1 (1)**
4	Cookies	25.4	3.6 (0.5)	2.4	3.0	1.4	1.9	0.1	0.7	0.3	0	0.5		0.7	0.8
5	Cow’s milk	14.9	12.2 (4.5) ^1^	**8.2 (4)**	**4.7 (5)**	**15.5 (3)**	**11.2 (3)**	**6.5 (4)**	**25.2 (2)**	**22.9 (2)**	**2.9 (4)**	**18.9 (2)**		0.6	**6.5 (4)**
6	Infant cereal	14.3	7.6 (2.6)	**4.3 (5)**	**5.0 (4)**	**5.2 (5)**	**2.5 (4)**	**12.6 (2)**	**7.5 (3)**	0.6	**8.7 (3)**	**5.5 (4)**		**24.0 (2)**	**6.2 (5)**
7	Vegetables	11.4	2.3 (1.1)	0.3	0.4	0.2	0	0.5	0.2	1.0	0.7	0.2		0.3	0.2
8	Grain-based mixed dishes	10.3	12.5 (7.5)	1.4	2.0	1.0	0.5	0.3	0.4	0	0	0.2		0	1.7
9	Table sugar	10.0	1.6 (1.2)	0.9	1.7	0	0	0	0.1	0	0	0.4		0.1	0
10	Fish	9.7	1.5 (0.9)	0.2	0	1.5	0.2	0.2	0.3	0.9	0	0.4		0.4	0.6
11	Eggs	9.7	1.3 (0.7)	0.3	0	0.9	0.5	0.2	0.7	0.9	0	0.1		0.4	0.7
12	Crackers	8.6	1.1 (0.4)	0.8	0.8	0.5	0.9	0.4	0.2	0	0	0.2		0.3	0.3
13	Bread	8.3	2.0 (1.1)	1.0	1.4	1.2	0.3	1.2	0.4	0.2	0	0.2		1.1	0.6
14	Sugar sweetened beverages	6.6	2.8 (2.8) ^2^	0.5	0.8	0.2	0.1	1.9	1.0	**1.3 (5)**	1.3	0.4		0.9	0.9
15	Fruits	5.4	3.1 (2.5)	0.5	0.9	0.2	0	0.3	0.1	0.3	1.5	0.1		0.3	0.2
16	Cakes	5.1	1.0 (0.6)	0.6	0.7	0.3	0.5	0.4	0.1	0.4	0	0.2		0.4	0.3
17	Noodles	5.1	0.7 (0.7)	0.5	0.5	0.4	0.4	0.7	0.1	0	0	0		0.2	0.3
18	White corn	4.3	1.4 (1.3)	0.7	1.2	0.6	0.1	0.2	0.1	0	0	0		0.1	0.4
19	Sausages/luncheon meats	4.3	1.4 (1.8)	0.5	0.1	1.0	0.9	0.2	0.2	0.3	0	0.1		1.3	0.8
20	Toddler/preschooler formula	3.7	3.0 (3.19	1.3	1.2	1.7	**1.3 (5)**	2.3	2.3	**1.6 (4)**	**2.1 (5)**	**2.9 (5)**		**2.2 (5)**	5.1
Total contribution of top 20 foods				97.6	97.6	97.0	97.2	96.5	98.8	98.0	99.8	99.1		98.4	97.5

^1^ includes 10.7% liquid milk and 89.3% milk powder. ^2^ includes 86% liquid beverages and 14% beverage powder. Grey shadow and bold words highlight the top-five food sources of nutrients.

**Table 5 nutrients-10-01730-t005:** Usual energy and nutrient intakes from food and beverages for Filipino toddlers aged 12–23.9 months (*n* = 714).

	Dietary Reference Intakes ^1^			Mean/Median Intake Percentiles						Inadequate/Excessive Reported Intake	
Nutrients	EAR/AMDR	AI/RNI	UL	10th	25th	Median	Mean ± SE	75th	90th	% <EAR/AMDR	% >AMDR/ >UL
**Macronutrients**											
Energy intake (kcal/day)	784 (EER)			444	560	717	771 ± 12	918	1159	-	-
Total fat (g/d)	-	-	-	11.3	17.5	23.8	25.6 ± 0.5	31.3	42	-	-
Saturated fat (g/d)	-	-	-	2.3	4.4	7.9	8.5 ± 0.2	11	14.6	-	-
Protein (g/d)	14.5	-	-	11.2	15	20.8	23.3 ± 0.4	28.8	38.6	23	-
Carbohydrate (g/d)	-	-	-	63	80	102	112 ± 2	132	169	-	-
Total sugars (g/d)	-	-	-	6	17	29	31.6 ± 0.8	41	59	-	-
Dietary fiber (g/d)	-	6–7	-	0.7	1.2	2	2.3 ± 0.1	3.1	4.5	-	-
**As percentage of total energy**											
Total Fat (%)	25–35 ^a^	-	-	18	24.1	30.3	29.6 ± 0.3	35.7	40.1	28	28
Protein (%)	6–15 ^a^	-	-	8.6	9.8	11.5	11.9 ± 0.1	13.6	15.6	<1	13
Carbohydrate (%)	50–69 ^a^	-	-	47.1	52	57.6	58.4 ± 0.3	64.3	70.9	18	13
**Antioxidants**											
Vitamin C (mg/d)	11.5	-	400	5.3	14.2	24.5	31.9 ± 1.1	37.7	69	21	0
Vitamin E (mg/d)	-	-	-	0.3	0.6	1.3	2.1 ± 0.1	2.7	5	-	-
**B vitamins**											
Thiamine (mg/d)	0.4	-	-	0.1	0.2	0.3	0.4 ± 0.01	0.6	0.9	59	-
Riboflavin (mg/d)	0.4		-	0.1	0.2	0.5	0.8 ± 0.03	1.1	1.9	42	-
Niacin (mg/d)	5	-	10	1.8	2.8	4.5	5.3 ± 0.1	7	10	57	10
Vitamin B6 (mg/d)	0.45	-	30	0.1	0.2	0.3	0.6 ± 0.05	0.6	1.2	61	<1
Folate (DFE µg/d)	120	-	-	28	47	81	223 ± 3	142	236	68	-
Vitamin B12 (µg/d)	0.85	-	-	0.3	0.4	0.8	1.1 ± 0.03	1.5	2.3	53	-
**Bone-related nutrients**											
Calcium (mg/d)	440	-	2500	121	179	321	479 ± 16	640	1058	62	<1
Phosphorus (mg/d)	380	-	3000	148	210	334	424 ± 11	551	826	57	0
Magnesium (mg/d)	-	60	65	22	32	48	57.1 ± 1.4	72	105	-	-
Vitamin D (µg/d)	-	*5*	50	0.1	0.5	1.3	2.3 ± 0.1	3	6.2	-	0
**Other micronutrients**											
Vitamin A (µg RE/d)	186.5	-	600	27	64	138	231 ± 10	307	554	60	8
Iron (mg/d)	6.7	-	-	1.5	2.3	3.6	4.9 ± 0.1	6.2	10.1	78	-
Zinc (mg/d)	2.7	-	7	1.3	1.7	2.6	4.2 ± 0.2	4.5	8.1	52	13
Sodium (mg/d)	-	*225*	-	189	288	438	517 ± 13	654	933	-	-
Potassium (mg/d)	-	*700*		231	322	453	539 ± 13	663	953	-	-
Selenium (µg/d)	13.3	-	90	12.8	18.2	26.1	30.5 ± 0.7	38	53.5	11	1

^1^ Philippine Dietary Reference Intakes 2015. EAR, estimated average requirements. AMDR, acceptable macronutrient distribution range. AI, adequate intake. RNI, reference nutrient intake. UL, tolerable upper intake level. DFE, dietary folate equivalents. RE, retinol equivalents.

**Table 6 nutrients-10-01730-t006:** Top-20 most consumed food groups and their contribution to energy and selected nutrient intakes among toddlers aged 12–23 months (*n* = 714) in the Philippines.

				Percent Contribution to Total Daily Intake (Ranking of Top 5 Food Sources of Nutrients)										
				Macronutrients				Vitamins				Minerals		
Rank	Food Groups	% of Children	Mean Intake (SE) Per Capita [g]	Energy	Carbohydrate	Protein	Total Fat	Thiamine	Riboflavin	Vitamin A	Vitamin C	Calcium	Iron	Zinc
1	Rice	91.2	62.3 (4.3)	**24.3 (1)**	**36.1 (1)**	**16.1 (1)**	1.1	**12.2 (3)**	**3.6 (4)**	0	0	**3.0 (4)**	**10.9 (3)**	**12.2 (3)**
2	Human milk	37.0	181.7 (3.4)	**15.9 (2)**	**12.1 (2)**	**10.9 (2)**	**25.1 (1)**	**4.2 (5)**	2.3	**3.5 (5)**	**23.1 (2)**	**9.7 (3)**	**11.3 (2)**	**9.4 (4)**
3	Cow’s milk	35.2	23.0 (2.2) ^1^	**12.7 (3)**	**6.8 (4)**	**10.9 (3)**	**20.9 (2)**	**13.0 (2)**	**43.4 (1)**	**36.9 (1)**	**8.4 (3)**	**35.7 (1)**	2.4	**11.0 (2)**
4	Fish	34.7	8.4 (0.6)	1.2	0	**7.5 (5)**	1.1	1.4	1.6	3.3	0	2.4	2.4	3.0
5	Vegetables	27.6	6.4 (0.6)	0.5	0.7	0.6	0.1	1.4	0.9	**4.9 (4)**	4.1	0.8	1.3	0.4
6	Cookies	24.8	4.9 (0.5)	**2.9 (5)**	**3.1 (5)**	1.4	2.9	1.3	0.9	0.6	0	0.7	1.5	0.8
7	Toddler/preschooler formula	22.8	23.3 (2.7)	**11.4 (4)**	**10.1 (3)**	**12.8 (4)**	**13.4 (3)**	**22.8 (1)**	**22.9 (2)**	**20.1 (2)**	**30.3 (1)**	**30.0 (2)**	**29.5 (1)**	**33.0 (1)**
8	Bread	21.1	6.0 (0.8)	2.5	3.2	2.7	0.9	3.4	1.2	0.3	0	0.6	3.8	1.6
9	Noodles	20.3	4.6 (0.9)	2.6	2.6	2.0	2.8	4.0	0.7	0.3	0.2	0.2	1.7	1.6
10	Eggs	19.9	5.4 (0.8)	1.0	0.1	2.9	2.1	1.0	**2.8 (5)**	3.1	0	0.4	2.0	2.2
11	Sugar sweetened beverages	19.6	14.3 (2.7) ^2^	2.1	3.0	0.9	0.7	**12.0 (4)**	**4.1 (3)**	**7.1 (3)**	**7.4 (4)**	**2.7 (5)**	**5.4 (4)**	**4.9 (5)**
12	Table sugar	19.3	3.1 (0.8)	1.6	2.7	0	0	0	0.2	0	0	0.8	0.2	0
13	Crackers	18.1	3.9 (0.6)	2.4	2.2	1.5	**3.2 (5)**	1.9	0.9	0	0	0.5	1.4	0.9
14	Fruits	14.4	9.2 (2.4)	1.0	1.6	0.4	0.2	0.8	0.5	0.8	**7.4 (5)**	0.8	1.2	0.4
15	Sausages/luncheon meats	12.7	5.6 (1.4)	2.0	0.3	3.6	**4.6 (4)**	1.2	1.1	1.2	0	0.4	**5.1 (5)**	3.2
16	Pork	9.7	2.7 (0.7)	1.0	0.1	1.8	2.6	2.4	0.8	2.9	0.1	0.2	0.9	2.6
17	Chicken	9.7	2.4 (0.7)	0.5	0	2.0	0.8	0.3	0.3	0.3	0.1	0.3	0.6	0.9
18	Cakes	9.4	3.6 (1.1)	1.7	1.8	0.9	1.9	1.5	0.5	1.4	0	0.8	1.7	0.8
19	Grain-based mixed dishes without meat	9.1	17.3 (5.8)	1.8	2.3	1.1	0.7	0.4	0.6	1.2	0	0.3	0	1.9
20	White corn	6.6	3.9 (1.6)	1.8	2.7	1.4	0.2	0.7	0.3	0	0	0.1	0.3	0.8
Total contribution of top 20 foods				90.9	91.5	81.4	85.3	85.9	89.6	87.9	81.1	90.4	83.6	91.6

^1^ includes 13.9% liquid milk and 86.1% milk powder. ^2^ includes 82.5% liquid beverages and 17.5% beverage powder. Grey shadow and bold words highlight the top-five food sources of nutrients.

**Table 7 nutrients-10-01730-t007:** Usual energy and nutrient intakes from food and beverages for Filipino toddlers aged 24–35.9 months (*n* = 727).

	Dietary Reference Intakes ^1^			Mean/Median Intake Percentiles						Inadequate/Excessive Reported Intake	
Nutrients	EAR/AMDR	AI/RNI	UL	10th	25th	Median	Mean ± SE	75th	90th	% < EAR/AMDR	%>AMDR/>UL
**Macronutrients**											
Energy intake (kcal/day)	962 (EER)			454	591	783	839 ± 13	1025	1294	-	-
Total fat (g/d)	-	-	-	8.1	12.8	20.3	23.3 ± 0.5	30.5	42.5	-	-
Saturated fat (g/d)	-	-	-	3	4.5	6.9	8.8 ± 0.3	10.7	16.3	-	-
Protein (g/d)	14.5	-	-	13.1	18	25	27.5 ± 0.5	34.3	44.8	14	-
Carbohydrate (g/d)	-	-	-	73	93	122	129 ± 2	157	196	-	-
Total sugars (g/d)	-	-	-	7	13	24	28.8 ± 0.8	39	57	-	-
Dietary fiber (g/d)	-	6–7	-	1.6	2.3	3.2	3.5 ± 0.1	4.4	6.5	-	-
**As percentage of total energy**											
Total Fat (%)	25–35 ^a^	-	-	12.7	17.4	23.1	23.4 ± 0.3	29.2	34.4	58	9
Protein (%)	6–15 ^a^	-	-	9.7	11.1	12.7	12.9 ± 0.1	14.6	16.5	<1	21
Carbohydrate (%)	50–69 ^a^	-	-	50.8	57.1	63.9	63.6 ± 0.4	70.4	76	9	30
**Antioxidants**											
Vitamin C (mg/d)	11.5	-	400	4.5	9	16	23.1 ± 0.8	29.4	49.8	35	0
Vitamin E (mg/d)	-	-	-	0.5	0.9	1.6	2 ± 0.1	2.7	4.1	-	-
**B vitamins**											
Thiamine (mg/d)	0.4	-	-	0.2	0.3	0.4	0.5 ± 0.01	0.6	0.9	46	-
Riboflavin (mg/d)	0.4		-	0.2	0.3	0.6	0.8 ± 0.03	1	1.7	35	-
Niacin (mg/d)	5	-	10	3.3	4.6	6.5	7.1 ± 0.1	9	11.1	30	18
Vitamin B6 (mg/d)	0.45	-	30	0.2	0.3	0.5	0.8 ± 0.04	0.9	1.6	40	0
Folate (DFE µg/d)	120	-	-	36	59	96	115 ± 3	150	216	63	-
Vitamin B12 (µg/d)	0.85	-	-	0.5	0.8	1.3	1.6 ± 0.04	2	3	29	-
**Bone-related nutrients**											
Calcium (mg/d)	440	-	2500	82	148	295	421 ± 14	565	936	66	<1
Phosphorus (mg/d)	380	-	3000	191	278	414	476 ± 10	605	838	44	0
Magnesium (mg/d)	-	60	65	33	46	64	71.5 ± 1.4	91	118	-	-
Vitamin D (µg/d)	-	5	-	0.3	0.7	1.6	2.3 ± 0.1	3	5	-	-
**Other micronutrients**											
Vitamin A (µg RE/d)	186.5	-	600	54	115	229	365 ± 18	430	772	41	15
Iron (mg/d)	6.7	-	-	1.9	2.7	4.2	5.2 ± 0.1	6.6	9.8	75	-
Zinc (mg/d)	2.7	-	7	1.3	1.9	2.9	5 ± 0.1	4.3	6.2	46	7
Sodium (mg/d)	-	225	-	252	381	572	646 ± 14	827	1130	-	-
Potassium (mg/d)	-	700		256	352	497	555 ± 11	693	926	-	-
Selenium (µg/d)	13.3	-	90	18	26	36	38.9 ± 0.7	59	64	3	2

^1^ Philippine Dietary Reference Intakes 2015. EAR, estimated average requirements. AMDR, acceptable macronutrient distribution range. AI, adequate intake. RNI, reference nutrient intake. UL, tolerable upper intake level. DFE, dietary folate equivalents. RE, retinol equivalents.

**Table 8 nutrients-10-01730-t008:** Top-20 most consumed food groups and their contribution to energy and selected nutrient intakes among toddlers aged 24–35.9 months (*n* = 727) in the Philippines.

					Percent contribution to total daily intake (ranking of top 5 food sources of nutrients)										
					Macronutrients				Vitamins				Minerals		
Rank	Food Groups	% of Children		Mean Intake Per Capita (g)	Energy	Carbohydrate	Protein	Total Fat	Thiamine	Riboflavin	Vitamin A	Vitamin C	Calcium	Iron	Zinc
1	Rice	96		76.8 (2.1)	33.1 (1)	47.6 (1)	20.9 (1)	1.8	15.3 (1)	5.0 (4)	0	0	5.2 (4)	15.7 (1)	18.5 (1)
2	Fish	50		16.9 (1.1)	2.3	0.1	12.8 (3)	2.6	2.5	3.2	5.8	0	5.5 (3)	4.4	4.7
3	Vegetables	39		13.9 (1.2)	1.2	1.6	1.0	0.3	2.4	1.5	8.3 (5)	11.7 (5)	1.9	3.0	1.1
4	Cow’s milk	38		25.4 (2.3) ^1^	12.6 (2)	6.4 (2)	19.4 (2)	25.0 (1)	13.0 (3)	45.9 (1)	34.8 (1)	11.8 (4)	43.4 (1)	2.3	11.7 (3)
5	Sugar sweetened beverages	33		29.8 (3.4) ^2^	3.6	4.9	1.3	1.4	16.6 (3)	7.2 (3)	10.1 (3)	23.6 (1)	5.1 (5)	8.7 (3)	7.9 (4)
6	Table sugar	29		3.3 (0.5)	1.6	2.5	0	0	0	0.2	0	0	0.9	0.2	0
7	Bread	28		12.0 (1.1)	4.9 (4)	5.9 (3)	4.7	2.2	6.0	2.3	1.0	0	1.5	7.7 (4)	3.3
8	Noodles	27		7.6 (0.9)	3.9 (5)	3.8 (5)	2.8	5.1 (5)	6.6 (5)	1.6	0.2	0.2	0.4	2.8	2.6
9	Eggs	25		5.3 (0.7)	1.3	3.1	3.5	3.3	1.1	4.0 (5)	3.9	0	0.6	2.6	3.0
10	Cookies	20		5.3 (0.7)	3.1	2.1	1.3	4.0	1.3	1.0	0.8	0	0.9	1.8	0.9
11	Fruits	17		11.0 (2.0)	1.1	1.7	0.4	0.2	1.2	0.7	0.7	17.0 (3)	0.8	1.5	0.5
12	Pork	16		7.4 (1.4)	2.7	0.1	4.7	8.2 (3)	5.7	2.9	12.1 (2)	0.6	0.4	2.8	7.2 (5)
13	Chicken	16		7.4 (1.4)	1.3	0	5.6 (5)	2.5	0.8	1.7	3.4	0.6	1.1	2.0	3.3
14	Toddler/preschooler formula	15		12.6 (2.2)	5.8 (3)	4.7 (4)	5.7 (4)	8.7 (2)	11.2 (4)	12.5 (2)	9.3 (4)	22.0 (2)	20.5 (2)	15.7 (2)	18.1 (2)
15	Sausages/luncheon meats	14		7.3 (1.4)	2.3	0.5	3.8	6.5 (4)	1.4	1.2	1.9	0	0.5	6.5 (5)	4.2
16	Crackers	14		3.5 (0.5)	2.1	1.8	1.2	3.4	1.5	0.8	0	0	0.5	1.3	0.8
17	Human milk	11		9.9 (0.4)	0.8	0.6	0.5	1.6	0.2	0.1	0.2	1.8	0.6	0.6	0.5
18	Cakes	11		4.5 (1.2)	2.1	2.1	1.0	2.7	1.7	0.5	1.7	0	1.0	2.1	0.9
19	Candy	10		2.2 (0.5)	1.3	1.3	0.5	1.6	0.3	0.4	0.2	0	1.0	0.7	1.0
20	Beans, nuts & peas	6		1.6 (1.4)	0.6	0.5	1.1	0.4	1.3	0.3	0	0.3	0.7	1.0	0.6
Total contribution of top 20 foods				87.7	91.3	92.2	81.5	90.1	93.0	94.4	89.6	92.5	83.4	90.8

^1^ includes 22.8% liquid milk and 77.2% milk powder. ^2^ includes 75.8% liquid beverages and 24.2% beverage powder. Grey shadow and bold words highlight the top-five food sources of nutrients.

**Table 9 nutrients-10-01730-t009:** Usual energy and nutrient intakes from food and beverages for Filipino young children aged 36–59.9 months (*n* = 2427).

	Dietary Reference Intakes ^1^			Mean/Median Intake Percentiles						Inadequate/Excessive Reported Intake	
Nutrients	EAR/AMDR	AI/RNI	UL	10th	25th	Median	Mean ± SE	75th	90th	% < EAR/AMDR	%>AMDR/>UL
**Macronutrients**											
Energy intake (kcal/day)	1119 (EER)			596	747	949	997 ± 7	1195	1458	-	-
Total fat (g/d)	-	-	-	8	13	21	24.1 ± 0.3	31	44	-	-
Saturated fat (g/d)	-	-	-	3	5	8	11 ± 0.2	13	21	-	-
Protein (g/d)	17.5	-	-	18	23	30	31.5 ± 0.3	38	48	10	-
Carbohydrate (g/d)	-	-	-	99	124	156	164 ± 1	195	237	-	-
Total sugars (g/d)	-	-	-	8	15	25	28.8 ± 0.4	38	54	-	-
Dietary fiber (g/d)	-	8–10	-	2.7	3.4	4.5	4.9 ± 0.04	5.9	7.4	-	-
**As percentage of total energy**											
Total Fat (%)	15–30 ^a^	-	-	10	15	20	20.2 ± 0.2	26	31	27	12
Protein (%)	6–15 ^a^	-	-	10	11	13	12.7 ± 0.04	14	15	0	13
Carbohydrate (%)	55–79 ^a^	-	-	55	61	68	67.1 ± 0.2	74	78	10	8
**Antioxidants**											
Vitamin C (mg/d)	17	-	650	5	8	14	16.9 ± 0.3	22	33	60	0
Vitamin E (mg/d)		5	-	0.9	1.4	2.1	2.4 ± 0.03	3	4.2	-	-
**B vitamins**											
Thiamine (mg/d)	0.45	-	-	0.25	0.35	0.49	0.54 ± 0.01	0.68	0.89	43	-
Riboflavin (mg/d)	0.45	-	-	0.22	0.32	0.51	0.61 ± 0.01	0.78	1.12	43	-
Niacin (mg/d)	5	-	15	4.8	6.3	8.4	8.92 ± 0.07	11	13.7	12	6
Vitamin B6 (mg/d)	0.5	-	40	0.36	0.47	0.7	1.3 ± 0.03	1.36	2.85	29	-
Folate (DFE µg/d)	160	-	400	55	80	119	123 ± 3	167	221	72	0
Vitamin B12 (µg/d)	0.95	-	-	1.03	1.39	1.94	2.1 ± 0.02	2.67	3.5	7	-
**Bone-related nutrients**											
Calcium (mg/d)	440	-	2500	107	156	239	286 ± 4	365	526	84	0
Phosphorus (mg/d)	405	-	3000	262	346	461	493 ± 4	604	763	38	0
Magnesium (mg/d)	-	70	-	51	66	85	91.1 ± 0.7	110	138	-	-
Vitamin D (µg/d)	-	5	-	0.9	1.3	1.9	2.2 ± 0.03	2.8	3.9	-	-
**Other micronutrients**											
Vitamin A (µg RE/d)	220	-	900	104	161	244	279 ± 3.5	356	493	43	1
Iron (mg/d)	7.5	-	-	2.4	3.6	5	5.6 ± 0.06	6.9	9.3	90	-
Zinc (mg/d)	3.25	-	12	1.9	2.5	3.4	4.8 ± 0.1	5.3	10.1	47	6
Sodium (mg/d)	-	300	-	319	470	681	740 ± 8	946	1238	-	-
Potassium (mg/d)	-	1400	-	366	465	607	647 ± 5	786	984	-	-
Selenium (µg/d)	15.85	-	150	28	37	49	51.7 ± 0.4	63	79	1	-

^1^ Philippine Dietary Reference Intakes 2015. EAR, estimated average requirements. AMDR, acceptable macronutrient distribution ranges. AI, adequate intake. RNI, reference nutrient intake. UL, tolerable upper intake level. DFE, dietary folate equivalents. RE, retinol equivalents.

**Table 10 nutrients-10-01730-t010:** Top-20 most consumed food groups and their contribution to energy and selected nutrient intakes among young children aged 36–59.9 months (*n* = 2427) in the Philippines.

				Percent Contribution to Total Daily Intake (Ranking of Top 5 Food Sources of Nutrients)										
				Macronutrients				Vitamins				Minerals		
Rank	Food Groups	% of Children	Mean Intake Per Capita (g)	Energy	Carbohydrate	Protein	Total Fat	Thiamine	Riboflavin	Vitamin A	Vitamin C	Calcium	Iron	Zinc
1	Rice	93.2	114.5 (0.6)	41.2 (1)	55.5 (1)	27.0 (1)	2.8	21.3 (1)	8.8 (3)	0.0	0.0	10.8 (3)	21.6 (1)	26.0 (1)
2	Fish	55.7	23.5 (0.4)	2.7	0.1	15.5 (2)	3.4	2.9	5.3 (5)	10.2 (4)	0.0	12.3 (2)	6.1 (5)	7.1 (4)
3	Vegetables	45.1	19.8 (0.6)	1.5	1.8	1.4	0.6	3.5	2.9	15.5 (3)	21.8 (3)	4.4 (5)	3.9	1.5
4	Sugar sweetened beverages	42.6	50.1 (1.9) ^1^	4.5 (4)	5.9 (3)	1.5	1.8	18.9 (2)	10.5 (2)	15.8 (2)	36.2 (1)	9.6 (4)	10.1 (2)	9.4 (3)
5	Cow’s milk	31.5	16.1 (0.8) ^2^	5.4 (2)	2.7 (5)	8.6 (3)	12.5 (2)	6.6 (4)	27.9 (1)	20.9 (1)	7.5 (4)	31.9 (1)	1.5	6.3 (5)
6	Table sugar	30.0	3.0 (0.1)	1.2	1.8	0.0	0.0	0.0	0.2	0.0	0.0	1.1	0.2	0.0
7	Bread	29.3	15.9 (0.6)	5.3 (3)	6.0 (2)	5.4	2.9	7.2 (5)	3.5	2.1	0.0	2.7	9.2 (4)	4.0
8	Eggs	26.5	9.8 (0.4)	1.4	0.1	4.0	4.3	1.4	6.1 (4)	6.4	0.0	1.2	3.1	3.8
9	Noodles	24.8	8.8 (0.4)	3.9 (5)	3.4 (4)	2.8	6.2 (4)	6.1	2.0	0.4	0.4	0.7	2.9	2.8
10	Pork	20.8	11.2 (0.7)	3.5	0.2	6.0 (5)	12.7 (1)	8.1 (3)	3.9	5.3	0.6	1.0	2.9	10.5 (2)
11	Sausages/luncheon meats	17.3	11.7 (0.8)	3.1	0.5	5.3	10.3 (3)	2.2	2.5	2.2	0.0	1.4	9.3 (3)	6.1
12	Cookies	17.2	5.4 (0.4)	2.6	2.4	1.1	4.2 (5)	1.2	1.1	1.0	0.0	1.3	1.7	0.9
13	Chicken	17.1	9.6 (0.6)	1.5	0.0	6.1 (4)	3.4	1.6	3.5	7.9 (5)	1.3 (5)	1.6	2.3	3.6
14	Fruits	16.7	14.3 (1.3)	1.3	1.7	0.5	0.4	1.4	1.1	1.3	23.6 (2)	1.3	1.8	0.7
15	Cakes	13.6	6.6 (0.6)	2.5	2.4	1.2	3.7	2.2	0.9	2.8	0.0	2.2	2.7	1.2
16	Crackers	12.6	3.7 (0.2)	1.8	1.5	1.1	3.5	1.5	1.0	0.0	0.0	0.8	1.2	0.8
17	Candy	9.7	1.9 (0.2)	0.9	0.8	0.5	1.5	0.3	0.5	0.3	0.0	1.4	0.6	0.9
18	Beans, nuts & peas	8.5	2.3 (0.4)	0.9	0.6	1.5	1.2	1.8	0.6	0.1	0.5	0.9	1.5	1.1
19	Savory snacks	8.2	2.0 (0.5)	0.8	0.7	0.3	1.6	0.3	0.2	0.2	0.0	0.6	0.8	0.6
20	Sweet biscuits	6.5	2.1 (0.2)	1.0	0.9	0.4	1.7	0.5	0.3	0.9	0.0	0.6	0.7	0.1
Total contribution of top 20 foods				87.1	89.1	90.2	78.6	88.9	82.9	93.4	91.9	88.0	84.2	87.2

^1^ includes 88.6% liquid beverages and 11.4% beverage powder. ^2^ includes 36.6% liquid milk and 63.4% milk powder. Grey shadow and bold words highlight the top-five food sources of nutrients.

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
