# Peer review of "Nutrient Intakes and Food Sources of Filipino Infants, Toddlers and Young Children are Inadequate: Findings from the National Nutrition Survey 2013"

_nutrients, 2018, doi:10.3390/nu10111730_

Reviewer 1 Report

Title: Nutrient Intakes and Food Sources of Filipino Infants, Toddlers and Young Children are Inadequate: Findings from the National Nutrition Survey 2013

In the present paper 4218 children, aged 6-59.9 months from the National Nutrition Survey 2013 were included. The purpose of this study was to describe the dietary intakes of infants, toddlers and young children aged from 6 months to 5 years.

The authors concluded that the intakes of total fat as percentage of energy and most of micronutrients were highly inadequate.

Reviewer’s comments;

This is a well-done study and authors used repeated 24-h dietary recall.

Line- 70: the abbreviation “FNRI” should be explained the first time

Line- 75: a second 24-h dietary recall 75 was repeated in 50% of randomly selected households on a non-consecutive day. Why and how this has been carried out should be explained more, and how this has been handled under the analysis should be elaborated more.

Line-269: Data on regions, rural, urban and socio-economic factors (education income etc.) were collected in this study. Do these factors effected the nutrient intakes and food sources of Filipino infants, toddlers and young children?

Line-2713; it seems that Selenium intake is very high in all infant and children groups. Please explain what is the sources of this high selenium intake and its risk?

Line-319: please discuss more about the limitation of this study including; the use of 24-h dietary recall calculating the micronutrient intake without including the use of supplements and other possible bias..

Author Response

Response to Reviewer 1 Comments

Point 1: This is a well-done study and authors used repeated 24-h dietary recall.

Response 1: Thank you very much for this feedback.

Point 2: Line- 70: the abbreviation “FNRI” should be explained the first time

Response 2: Apology for this. Food and Nutrition Research Institute (FNRI) is added at Line-71.

Point 3: Line- 75: a second 24-h dietary recall was repeated in 50% of randomly selected households on a non-consecutive day. Why and how this has been carried out should be explained more, and how this has been handled under the analysis should be elaborated more.

Response 3:

More information is added as below.

Lines 77-80: “…… To estimate the day-to-day variance component in energy and nutrient intake required for usual intake analysis, first 24-h dietary recall was collected for all children and a second 24-h dietary recall was repeated in 50% of randomly selected households on a non-consecutive day. The second 24-h dietary recall was typically collected two days after the first 24-h recall.”

Lines 137-142: “……This program estimates distributions (in percentiles) of usual nutrient intake by removing the effect of day-to-day (intra-person) variability in intake from daily intakes, as well as the proportion below estimated average requirements (EAR) defined by the Philippine Dietary Reference Intakes 2015 [24]. Hence, the prevalence of inadequacy in the population is estimated as the proportion of individuals with usual intakes below the EAR [25].”
Point 4: Line-269: Data on regions, rural, urban and socio-economic factors (education income etc.) were collected in this study. Do these factors effected the nutrient intakes and food sources of Filipino infants, toddlers and young children?

Response 4: Indeed, these factors did have effects on nutrient intakes and food sources.  Because the amount of data that is described in this manuscript is already larger, the effects of place and of residence and SES on food and nutrient intakes will be presented and discussed in our next manuscript, which is already planned.  For now, some data on the effects of these factors on nutrient intakes and food consumption can be found in the Report of 8th National Nutrition Survey Dietary Survey in FNRI website (http://www.fnri.dost.gov.ph/images/facts_and_figures/Dietary-2013.pdf). Example findings include: (1) In urban households, consumption of cereal products (mainly bakery products), meat and meat products, poultry, fruits, milk and milk products, miscellaneous food (particularly beverage) and eggs were higher than in rural households where rice and rice products, corn and corn products, fish and fish products, and vegetables were consumed in greater quantity. (2) The percent contribution of rice and rice products to energy, protein, iron, thiamin, riboflavin and niacin was highest among the poorest households. The percent contribution to energy decreases as the wealth quintile increases. Fish and fish products were the animal sources of protein among the poor and poorest households while poultry and meat contributed more to protein intake among the rich and richest households. (3) The percent contribution of fruits to vitamin C intake was highest among households in the richest quintile.

Point 5: Line-273; it seems that Selenium intake is very high in all infant and children groups. Please explain what is the sources of this high selenium intake and its risk?

Response 4: Please see attached table for food sources of selenium in this study population. Before 12 months, human milk, rice, and rice gruel (rice product) were the top three sources of selenium and after 12 months, rice, fish and bread are the top three food sources. Fish, rich in selenium, was one of the commonly consumed foods. Selenium content in rice used in this study was 15.2µg per 100 grams. Rice, consumed in a high amount in this population. For example, mean intake of rice among 36-59.9 month olds was 114.5g, therefore, provided 17.0 µg selenium. We think the relatively high consumption of rice and fish consumed resulted a relatively high selenium intake in this population.

Regarding risk of high selenium intake, the tolerable upper intake levels for selenium defined by Philippine Dietary Reference Intakes are 60 µg for 6-11 months, 90 µg for 12-23.9 months and 90-150 µg for 36-59.9 months. Based on these levels, 0-2% the children had selenium intake above the ULs. Therefore, we do not think the risk for high selenium intake is a concern.

Point 6: Line-319: please discuss more about the limitation of this study including; the use of 24-h dietary recall calculating the micronutrient intake without including the use of supplements and other possible bias.

Response 6:

More limitations are discussed in Lines 332-346:

“This study has several limitations. First, we examined intake on a given day, therefore, we may have underestimated the consumption of foods that are not consumed on a daily basis. Second, this study relied on mother or caregiver reports of child intake. The participants may have over-or under-estimated their child’s consumption during the recall [37]. However, because of the large sample size with corresponding survey weights applied in all the datasets, these limitations could be overshadowed and could represent valuable national information. Third, because the purpose of this study was to estimate nutrient intakes from food and beverages only, the use of dietary supplements were not included. This could lead to underestimation of total daily intakes of micronutrients in these children. A further study that assesses nutrient intakes from foods, beverages and dietary supplements is warranted. Last, about half of the data in the food composition database was built by adopting data from the National Nutrient Database of United States Department of Agriculture and from some other nearby countries. Although efforts were made to select the foods that could match the equivalent foods in the Philippines, food composition of foods from other countries may not reflect the foods in the Philippines. This could also induce over-or underestimation of nutrient intakes in this population.”

Reviewer 2 Report

This manuscript describes the intakes of foods and nutrients in Filipino children between 6 months and 5 years of age.  The data collection is from a national survey and includes some 4000 infants and children, providing important data of nutritional intake in this group.

Small edits are noted in an attached manuscript, particularly in regard to defining abbreviations and clarifying the most consumed foods classification.

This data is potentially useful for people in the Philippines to inform campaigns encouraging mothers to provide young children with more nutrient dense foods.  Further work may be necessary to understand why children are offered the foods reported here.  Is it custom, lack of knowledge on the part of the mothers/caregivers, or cost of meat and vegetables for example.

The main limitation of this paper for publication in an international journal is the lack of relevance for other countries.  The discussion makes no attempt to fit the findings in the broader context of early childhood nutrition in other countries.

Author Response

Response to Reviewer 2 Comments

Point 1: Small edits are noted in an attached manuscript, particularly in regard to defining abbreviations and clarifying the most consumed foods classification.

Response 1:

Sorry for missing full name of several abbreviations including FNRI, USDA, FAO and INFOODS. All the full names were added before the abbreviations. In addition, a native English speaker checked and edited English of the manuscript.

Point 2: This data is potentially useful for people in the Philippines to inform campaigns encouraging mothers to provide young children with more nutrient dense foods.  Further work may be necessary to understand why children are offered the foods reported here.  Is it custom, lack of knowledge on the part of the mothers/caregivers, or cost of meat and vegetables for example.

Response 2:

Indeed, further work to understand the food consumption patterns are needed. Food choices are multi-dimensional phenomena. Custom, food availability, economic access to food and knowledge can all influence food consumption. For now, some data on the effects of place of residence and SES on food consumption have been reported in FNRI website: 8th National Nutrition Survey Dietary Survey (http://www.fnri.dost.gov.ph/images/facts_and_figures/Dietary-2013.pdf). Example findings include: (1) In urban households, consumption of cereal products (mainly bakery products), meat and meat products, poultry, fruits, milk and milk products, miscellaneous food (particularly beverages) and eggs were higher than in rural households where rice and rice products, corn and corn products, fish and fish products, and vegetables were consumed in greater quantity. (2) The percent contribution of rice and rice products to energy, protein, iron, thiamin, riboflavin and niacin was highest among the poorest households. The percent contribution of rice to energy decreases as the wealth quintile increases. Among animal food sources, fish and fish products were the sources of protein in the poor and poorest households while poultry and meat contributed more to protein intake in the rich and richest households. (3) The percent contribution of fruits to vitamin C intake was highest among households in the richest quintile.

Point 3: The main limitation of this paper for publication in an international journal is the lack of relevance for other countries.  The discussion makes no attempt to fit the findings in the broader context of early childhood nutrition in other countries.

Response 3:

This is a very valuable comment. We did think about the potential implication of this study to other high rice-consumption countries in Southeast Asia such as Indonesia, Vietnam and Cambodia where malnutrition of children is also prevalent but we did not think to include the point in the manuscript.

Now, additional text is added in Introduction, lines-54-55: “… and surrounding countries in Southeast Asia, where the prevalence of malnutrition is high [12]”. Additional text is also added in the first paragraph of Discussion, lines 273-277: “To our knowledge, this is the first study to provide current and comprehensive estimates of usual intakes of nutrients and food sources of key nutrients in children under-five year olds in the Philippines. In addition, the finding of this study could also provide insights to plan similar studies in other countries in Southeast Asia, where the diets of children are also largely based on rice and malnutrition remains a pressing issue.”

Round  2

Reviewer 2 Report

I think it would be useful to mention briefly the next steps that come after this work as discussed in the response 2 to reviewer 1. 

Author Response

Response to Reviewer 2 Comments

I think it would be useful to mention briefly the next steps that come after this work as discussed in the response 2.
Response:

Thanks for this comment. Brief discussion for next steps after this work (highlighted in green color) is added in Conclusions, Lines 356-360: “…….Further studies to understand factors influencing the dietary intake of this population, such as socioeconomic status of families, food consumption habits and food access in different resigns are underway. In addition, the findings and insights obtained from this study could be shared with the nutrition researchers and healthcare professionals in other countries in Southeast Asia, where malnutrition of children is prevalent.”
